# Bibliometric Analysis of the Informal Caregiver’s Scientific Production

**DOI:** 10.3390/jpm12010061

**Published:** 2022-01-06

**Authors:** Bruno Ferreira, Ana Diz, Paulo Silva, Luís Sousa, Lara Pinho, César Fonseca, Manuel Lopes

**Affiliations:** 1Hospital Beatriz Ângelo, 2674-514 Loures, Portugal; 2Centro Hospitalar de Setúbal, EPE, 2900-182 Setubal, Portugal; ananfdiz@gmail.com; 3Unidade Local de Saúde do Baixo Alentejo, EPE, 7801-849 Beja, Portugal; paulocesarlopessilva@gmail.com; 4São João de Deus School of Nursing, University of Évora, 7000-811 Evora, Portugal; lmms@uevora.pt (L.S.); lmgp@uevora.pt (L.P.); cfonseca@uevora.pt (C.F.); mjl@uevora.pt (M.L.); 5Comprehensive Health Research Centre (CHRC), 7000-811 Evora, Portugal

**Keywords:** informal caregiver, caregiver, bibliometrics, data analysis

## Abstract

(1) Background: Due to the increase in care needs, especially in the elderly, the concept of caregiver has emerged. This concept has undergone changes over the years due to new approaches and new research in the area. It is in this context that the concept of informal caregiver emerged. (2) Objectives: To analyse the evolution of the caregiver concept. (3) Methods: Bibliometric analysis, data collection (Web of Science Core Collection) and analysis (Excel; CiteSpace; VOSviewer). (4) Results: Obtained 22,326 articles. The concept emerged in 1990, being subjected to changes, mostly using the term “informal caregiver” since 2016, frequently related to the areas of Gerontology and Nursing. The following research boundaries emerged from the analysis: “Alzheimer’s Disease”, “Elderly” and “Institutionalization”. (5) Conclusions: The informal caregiver emerges as a useful care partner, being increasingly studied by the scientific community, particularly in the last 5 years. Registration number from Open Science Framework: osf.io/84e5v.

## 1. Introduction

The average life expectancy at birth, worldwide, increased from 66.8 years in 2000 to 73.3 years in 2021; there was also an increase in the average healthy life expectancy in the same period of time, from 58.3 years to 63.7 years respectively [1]. Countries with lesser economic power have seen a growth of about 11 years in average life expectancy at birth in the period between 2000 and 2016 [2], but between 2015 and 2019 the speed of this growth slowed. The African region had an average life expectancy of 64.5 years in 2019 and an average healthy life expectancy of 56 years. Europe had presented an average life expectancy in 2019 of 78.2 years and 68.3 years for average healthy life expectancy [1]. In Portugal, in the triennium 2017−2019, the average life expectancy at birth was 80.93 years, while the ageing index was 163.2 elderly per 100 young people in the year 2019, and this index is expected to double by 2080 [3]. Maintaining the quality of life of the elderly population has proved to be a pressing challenge, a consequence of the increase in the number of years with some degree of dependency, allied to the need for support or differentiated care, inherent to the increase in average life expectancy [4].

The aging process brings with it an increase in the degree of dependence of the elderly person, requiring care which brings family changes [4], which lead to the need to adapt to the new life situation. In this readjustment process, the specialist nurse in rehabilitation nursing is a facilitator in the management of the transition to the new reality, implementing interventions aimed not only at the individual, but also at the family/caregivers [5,6].

Within the scope of the caregiver’s role, a transition which is commonly seen is the transition of roles or transition of functions. In this type of transition there is a change in the role played, often within the family, which generates a reformulation of expectations and changes in the definition of agents as individuals and their role in the social context [5]. This transition requires an adaptation that includes the acquisition of knowledge, skills and abilities to deal with the problems that arise at the self-care level and that affect the well-being and, in this sense, the nurse plays a key role in promoting the person’s functional readaptation [7].

As the better positioned professionals to help the person in these transitions, nurses assess psychosocial needs and direct their intervention according to individual needs, empowering the informal caregiver to perform their role, allowing for greater adherence to the therapeutic regimen of the person in a situation of chronic disease [5,8].

The informal caregiver presents himself as an ally of the health team [8]. Person-centred care is a partnership between the person and their caregivers, whether family, neighbours or friends, who provide emotional, physical or practical support in response to illness, disability or age-related need [9]. This type of approach implies that the specific needs, preferences and expectations of the person and their caregivers are continuously assessed, respected and considered in care planning, implementation and adaptation of care over time [10,11].

Based on the approaches mentioned above, informal caregivers should be seen as an integral part of the health system, due to their high relevance in today’s society, in order to maintain the quality of care and quality of life of both the person receiving care and their own. The care developed should be centred on the person and the caregiver, who is considered in several countries as a critical component of high-quality care, respecting their needs, preferences and expectations when implementing and adapting the care plan for adherence to the therapeutic regime [12,13]. In this sense, the caregiver is the main focus of attention, which justifies the development of this research.

In the preparation of this research, it was found that there are several terms related to the informal caregiver/family caregiver and not all terms are part of the Descriptors in Health Sciences (DeCS) and Medical Subject Headings (MESH). On the other hand, it becomes important to identify the researchers who stand out for the quality and peer recognition of their contributions, as well as the pattern of publications and their behaviour over time. Thus, with this bibliometric analysis of published studies related to the concept of informal caregiver/family caregiver, we intended to answer the following research question: How is the scientific production on the concept of informal caregiver characterized? The general objective was to present the characteristics of the scientific production on the topic, specifically regarding the aspects of authorship, cited work, co-authorship networks and bibliographic coupling.

## 2. Materials and Methods

### 2.1. Type of Research

This is a bibliometric analysis, which essentially consists of a statistical and quantitative technique that allows the measuring of the production and dissemination rates of knowledge, monitoring of the development of the concept under study in the various scientific areas and areas of interest, and analysis of publication patterns [14]. This type of analysis aims to quantify and analyse the processes of written communication through mathematical and statistical analysis to determine the nature and historical evolution of a particular discipline or concept [15].

This analysis assumes significant importance since it deciphers and maps scientific knowledge as well as established nuances, responding to large volumes of data in a consistent manner, building solid bases for scientific advancement, identifying gaps in knowledge and positioning scientific contributions according to their respective fields of action. Thus, bibliometric analysis can reveal emerging development areas of scientific knowledge in a respective field, contributing to the evolution of scientific knowledge and generating research opportunities [16,17].

This methodology has been a rapidly growing academic interest among researchers due to the increased availability of databases and software used [16,18].

This analysis was conducted following four steps: (1) Definition of the study objective; (2) Choice of the analysis technique; (3) Collection of data for analysis; (4) Performance of the bibliometric analysis and exposure of the respective results [16].

### 2.2. Research Strategy, Study Selection and Data Extraction

For data extraction, a previous search was conducted in the Virtual Health Library (VHL), and the following DeCS/MeSH descriptors were found: “Care Giver”, “Care Givers”, “Caregiver”, “Caregiver, Family”, “Caregiver, Spouse”, “Caregivers, Family”, “Caregivers, Spouse”, “Family Caregiver”, “Family Caregivers”, “Spouse Caregivers”, “Informal Caregiver”. Criteria for their selection were also established, and included review and empirical studies, publications in newspapers and scientific journals in article format, with no chronological limit of publication, published in English, Spanish, French and Portuguese and which addressed or referred to the informal caregiver or any of the descriptors found. A search in Web of Science (Clarivate Analytics) was then performed with the following search equation: AK = ((Care Giver) OR (Care Givers) OR (Caregiver) OR (Caregiver, Family) OR (Caregiver, Spouse) OR (Caregivers, Family) OR (Caregivers, Spouse) OR (Family Caregiver) OR (Family Caregivers) OR (Spouse Caregivers) OR (informal Caregiver)) OR KP = ((Care Giver) OR (Care Givers) OR (Caregiver) OR (Caregiver, Family) OR (Caregiver, Spouse) OR (Caregivers, Family) OR (Caregivers, Spouse) OR (Family Caregiver) OR (Family Caregivers) OR (Spouse Caregiver) OR (Spouse Caregivers) OR (informal Caregiver)), refined by: Document Types: (ARTICLE) and Languages: (ENGLISH OR SPANISH OR FRENCH OR PORTUGUESE). Indexes: SCI-EXPANDED, SSCI, A&HCI, CPCI-S, CPCI-SSH, ESCI, CCR-EXPANDED, IC.

The search was performed on 25 June 2021 at 17:33 GMT+1, reserving any bias arising from the constant updating of the databases. After the search in the Web of Science, data collection began, and this was carried out in the form of a file compatible with the bibliometric analysis software used, these being Excel 2020 (Redmond, WA, USA), CiteSpace 5.7.R2 (Drexel University, Philadelphia, PA, USA), and VOSviewer (Leiden University, The Netherlands).

The search and data extraction were validated by three independent researchers, and in occasional cases, with any doubt or inconsistency of results, these were analysed by two other researchers, and discussed by the team of researchers in order to correct and eliminate duplicate data.

### 2.3. Statistical Analysis of the Data

The bibliometric analysis was carried out simultaneously to the extraction of the bibliometric data and took into consideration its two main techniques: performance analysis, which allows examining the contributions of research to a given field in a descriptive way and which reveals itself as the initial phase of bibliometric studies; and scientific mapping, which examines the relationships between the constituents of research, where analysis relates to the intellectual interaction and structural connections between the constituents of research [16].

The description of the data is based on graphs, figures and tables that were produced by analysis software such as VOSviewer [16], CiteSpace and Excel.

## 3. Results

### 3.1. Trend in the Annual Evolution of Publications

After applying the inclusion criteria described above, a sample of 22,326 articles was obtained. Since 1990, when the first articles on the subject were published, the number of publications has been increasing annually, with a more accentuated growth being noted from 2006 (Figure 1). Of the 22,326 articles, 1055 articles were published in the year 2021, up to the date of this research. This shows a growing interest in the subject, which has been increasing every year, without there ever having been a decrease in the number of publications from 2006 to the present day.

### 3.2. Distribution of Articles by Journal and Area of Publication

The 25 scientific journals or magazines with the highest number of published articles (*n* = 5130) were identified, corresponding to 22.98% of the total. The top three are: Aging Mental Health (*n* = 388) (1.74%), with a 2020 impact factor (2020 IF) of 3.66; Gerontologist (*n* = 358) (1.6%), with a 2020 IF of 5.27 and International Journal of Geriatric Psychiatry (*n* = 321) (1.44%), with a 2020 IF of 12.38. It is noteworthy that the Journal of Clinical Nursing appears in sixth place (*n* = 254) (1.14%), with an IF 2020 of 16.92, and in fifteenth place is the journal Disability and Rehabilitation, with (*n* = 162) (0.73%) and IF 2020 of 14.88.

Using the data from CiteSpace and VOSViewer, it is possible to identify the 25 research areas with the highest number of publications identified (Figure 2); the 10 with the highest number of publications on the topic are, in descending order: Geriatrics and Gerontology; Nursing; Psychology; Psychiatry; Health and Social Services; Public, Occupational and Environmental Health; Neurology and Neurosciences; Rehabilitation; Oncology; General and Family Medicine.

### 3.3. Distribution of Articles by Language and Country of Publication

With regard to the language chosen for publication and taking into account the selected languages for data extraction, the prevalence of the English language is evident, for 97% of the articles published (*n* = 21,664), with the remaining 3% represented by Spanish (*n* = 345), French (*n* = 194) and Portuguese (*n* = 123).

When analysing the data collected by VOSViewer and CiteCpace, and concerning the countries of origin of the publications (Figure 3), the United States of America (USA) appears with the largest presentation of publications (40.4%), followed by England, Canada, Australia and Spain, with Portugal occupying the 25th place (1.1%).

### 3.4. Authors Profile

Regarding scientific contribution to the subject of informal caregivers, the following authors stand out in descending order as those with more scientific contributions: Richard Schulz followed by Laura Gitlin, Steven Zarit, George Demiris, Debra Parker Oliver, among others, as can be verified in Figure 4.

### 3.5. Network and Density of Co-Citations by Author

Regarding the co-citation density of authors, this is represented by colour density, demonstrating the incidence of relevant authors. The hotter the colour (yellow) the greater the number of publications and citations of the author; at the other extreme, the colder the colour (blue) the lower the number of publications and citations of the author [19]. Thus, through the interpretation of the density network, a greater expressiveness of authors such as: Zarit; Folstein; Pearlin; Radloff and Schulz can be verified (Figure 5).

Regarding the network of author co-citations, this concerns documents in which authors are jointly cited, allowing understanding of grouping of authorship content [17]. The larger the cluster, the greater the number of citations that the author received; the same colour reflects a cooperative relationship between authors [19]; the closer the clusters, the stronger the relationship between them [17]. Taking such evidence as support and using the data in Table 1 and Figure 6, it was possible to identify 10 clusters. The first cluster is formed by 20 authors, highlighting Zarit with 1449 citations. The second cluster is formed by 12 authors, highlighting Folstein with 1397 citations. The third cluster with the most citations also stands out, with Pearlin appearing at its top with 1296 citations. The cluster led by Schulz, despite visually representing a cluster of larger dimensions, appears in fifth place with 977 citations and 282 authors.

The degree of centrality refers to the number of established ties that an author has in a co-authorship network [16]. Thus, Zarit, presents himself as the author with the highest centrality, followed by Folstein and Perlin (Table 1).

### 3.6. Network and Density of Bibliographic Coupling

Bibliographic coupling represents a scientific mapping technique and is associated with the relationships between citing publications and themes. Assuming that two publications that share the same bibliographic references are similar in their content, bibliographic coupling divides the publications into thematic groups based on the shared references [16].

Thus, regarding bibliographic coupling, it was possible to determine, through the interpretation of the graphic representations produced by the VosViewer tool, the most cited articles, these being “2013 Alzheimer’s disease facts and figures” by Thies (2013); “Cancer and caregiving: the impact on the caregiver’s health” by Nijboer et al. (1998); “Differences between caregivers and non-caregivers in psychological health and physical health: A meta-analysis” by Pinquart (2011); “Introduction to the special section on Resources for Enhancing Alzheimer’s Caregiver Health (REACH)” by Gitlin et al. (2003); and “Predicting Caregiver Burden and Depression in Alzheimer’s Disease” by Clyburn et al. (2000) (Figure 7).

It was also possible to determine the clusters of articles that cite each other (Figure 8), as well as the temporal evolution of these same citations (Figure 9). Thus, we were able to trace some networks of links between references in order to highlight them: Thies (2013) appears with a cluster of large dimensions and in isolation from the other references; Nijboer (1998) appears with a cluster and very close to Nijboer (1999); Pinquart (2011) also appears with a cluster of significant dimensions with some proximity to Pinquart (2004); Gitlin (2003) appears in isolation and Clyburn (2000) appears with some proximity to Seltzer (2000) (Figure 8).

By date, it can be gauged that the greatest density of scientific production occurs from 2005 onwards, with Thies (2013) being a work of great relevance, followed by Pinquart (2011) (Figure 9).

### 3.7. Analysis of Bursts

Using CiteSpace IV, it was possible to analyse the most commonly used keywords in citations in the period between 1995 and 2025, most relevant being: Alzheimer’s disease, family caregiver, elderly, institutionalization and stress (Table 2). The same process was carried out to ascertain the references with stronger citation bursts, with George (1986); Braun (2006); Stone (1987); Adelman (2014); Cantor (1983) standing out in these (Table 3).

### 3.8. Keyword Network and Co-Occurrence Density

The interpretation of the graphical representations obtained by the VosViewer allows the co-occurrence density of the keywords to be ascertained through the evaluation of the colour of the density diagram, keywords which appear most frequently being perceptible. The higher the occurrence of the keywords, the warmer the colour becomes (yellow) and the lower the occurrence, the colder the colour becomes (blue) [19]. Thus, it could be perceived that the keywords that appear with greater frequency are: caregivers; family caregivers; dementia; burden and depression (Figure 10). In the following figure (Figure 11), we were able to interpret the co-occurrence network of the key words, where the larger the cluster, the greater the frequency with which the key word appears, and the colour of the clusters represents the relationship between the key words: family caregivers, dementia and burden may be associated to the same cluster, while caregivers and depression are found in distinct clusters.

The chronology of occurrence of these keywords can also be inferred (Figure 12), with the terms caregivers and family caregivers appearing in 2013 and informal caregivers in 2016, and no new terms appearing after 2016. In 2013, the terms burden and depression appeared in addition to the terms caregivers and family caregivers; in 2016, the terms anxiety, experiences and oncology appeared in addition to the term informal caregivers.

### 3.9. Analysis of Keyword Clusters

Through Citespace it was possible to carry out a network analysis of clusters by keywords. This procedure will allow us to verify if the works under analysis have been homogeneous in the use of descriptors. The Silhouette score is an indicator of the homogeneity or consistency of the cluster under analysis and Silhouette score values close to 1 confirm this homogeneity [20]. The largest keyword clusters present a Silhouette score above 0.77, revealing moderate to high consistency and homogeneity (Table 4).

## 4. Discussion

In this study, a bibliometric analysis of global trends on the concept of informal/family caregiver was conducted. The timeframe was determined based on the first publication on the topic (1990) until today (2021).

The analysis shows a steady increase in the number of publications over the years.

English is the most widely used language and the United States of America is the country that has contributed the most to research on the subject, with Portugal occupying 25th place in terms of number of publications.

After data analysis and as previously mentioned, it is possible to state that the use of the term informal caregiver began in 1990, with an exponential and constant growth from 2006 onwards, an evolving use of the term being currently perceptible.

Among the 25 scientific journals with the largest number of publications, the following journals stand out: “Aging and Mental Health”, “Gerontologist” and “International Journal of Geriatric Psychiatry”, with impact factors of 3.66, 5.27 and 12.38, respectively.

The largest number of publications appears in journals or magazines dedicated to Gerontology & Geriatrics, Nursing and Psychology. The largest number of co-citations belongs to Schulz, Gitlin and Zarit, the latter being the author the author with the most published articles on the topic, followed by Folstein and Pearlin, Alzheimer’s disease”, “institutionalization” and “family caregiver” appearing as the keywords with the strongest citation bursts. With the support of Table 2, it is possible to indicate that these keywords appear with the strongest citation bursts since the beginning of the 1990s until currently, suggesting that they are themes still discussed by the authors. Eom & Fortunato (2011) [21], when studying the dynamic properties of citation flows, found that the first years after the publication of articles are characterized by citations bursts that work as an indicator of the popularity dynamics of the evidence produced and that have different durability over time.

The most commonly used terms in the publications are “caregivers” and “family caregivers”, which appeared in 2013, with the term “informal caregivers” gaining greater relevance from 2016 onwards, maintaining its expression to date.

Given this lexometric change, and in order to avoid ambiguity in the applicability of the concept, it will be important to analyze it in a future study. According to Tofthagen & Fagerstrøm (2010) [22], nursing research should focus on the unambiguous use of concepts, for which Rodgers’ method is a possible method.

### 4.1. Research Frontiers

The citation bursts of the keywords were identified using CiteSpace, in order to predict research frontiers. Through the analysis of Table 2, it is concluded that the keywords with strongest citation bursts are “Alzheimer’s Disease”, “Elderly” and “Institutionalization”, thus revealing the most relevant research frontiers.

### 4.2. Alzheimer’s Disease

Alzheimer’s disease is a progressive, disabling and long-term neurodegenerative disease, characterised by cognitive impairment, progressive loss of autonomy and behavioural disorders, often manifesting with memory loss and subsequent progression to an inability to perform basic activities of daily living [23,24,25,26,27,28].

By 2050, an increasing incidence of dementias (including Alzheimer’s disease) is expected to have tripled from 2010, with a consequent increase in the number of informal caregivers [24].

The informal caregiver is the person who provides more time to monitor and meet the needs of the person with Alzheimer’s disease, in order to provide the continuity of a life with dignity, often being performed by a close relative (spouses, children), an unpaid role in most situations [27]. In addition to the challenges of their own daily lives, the informal caregiver is subject to a high risk of exposure to situations of chronic stress, with consequent impact at physical, emotional, social and financial levels, being associated with feelings of isolation, anxiety and depressive symptoms, an increased risk of cardiovascular disease, decreased immunity and increased mortality being described in the literature. For this reason, they are denominated as the secondary victims of this disease [21,25].

The ability of informal caregivers to cope with the recurrence of demands influences the quality of care to the person [28]. Lack of strategies to cope with these demands may have a negative impact on the quality of care provided to the person as well as on the quality of life of the informal caregivers themselves. The literature shows that the quality of life of informal caregivers of people with Alzheimer’s disease has been shown to be lower than that of informal caregivers of people without Alzheimer’s disease, which may be one of the driving factors for the abandonment of care provision, as well as the increase in work restrictions and decrease in productivity rates [24,26].

### 4.3. Elderly

According to the latest United Nations report on population prospects, the number of people aged 65 years and overrepresented 9% of the world population in 2019, and is expected to reach 16% by 2050, where the number of people aged 80 years and over will be about three times as high [29]. Of this 16%, about half, will develop some kind of disability or limitation that will undoubtedly require assistance from a responsible person, whereby the informal caregiver assumes a preponderant role in the discharge of needs associated with ageing [30].

The principle of aging as well as the process of aging constitute a positive phenomenon at individual and collective level, proving the progress achieved at economic, social and biomedical levels, constituting, utopically, the favourable culmination of human development, translating into gains inherent to longevity, but bringing alongside it increased responsibilities regarding the response to the needs of the frail person by the degenerative process inherent to aging. It is then possible to affirm that the loss of valences, namely physical and mental, can determine the vulnerability and frailty of the elderly person and compromise the success of this same aging process [31,32].

The increase not only in the longevity index but also in the dependency index verified in the last decades [33] poses new challenges at the social level, necessitating the development of care provision models centred on the elderly person and his/her caregiver, contemplating care of excellence and safety for both parties involved [32]. Thus, physical frailty is defined as a consequence of multiple causes inherent to the aging process and characterized by a decrease in strength and resistance, reduction of physiological function, consequent increase of vulnerability and subsequent development of dependence [9,30]. As previously mentioned, family caregivers are mostly family members, partners, spouses, friends or neighbours who assist in a wide range of care assistance to the frail elderly person, experiencing a substantial physical, financial and psychosocial burden, as well as stress associated with decreased quality of life by the continuous character and longevity of their role as caregivers, often going against the satisfaction of their own needs [24,29,30]. Currently, musculoskeletal symptoms are closely related to the provision of care by informal caregivers, influenced by emotional factors, excessive workloads and poor training, particularly at the ergonomic level [29].

### 4.4. Institutionalization

Increased longevity inevitably associated with increased incidence of morbidities is one of the predominant factors justifying institutionalization [29].

With the ageing of population and the increasing prevalence of chronic diseases affecting all age groups, the integration of home care services is becoming a necessity for frontline health service organizations worldwide [9].

Since hospitalization is one of the key factors in the increased costs associated with the use of health services related to chronic diseases, it is essential to implement effective and safe alternatives to conventional hospitalization [9]. The overload of emergency services and subsequent hospitalizations are closely related to increased risks for the elderly population, namely functional and cognitive imbalance, as well as loss of independence [9]. Home hospitalization emerges as an alternative to providing care in a hospital environment for people who would be institutionalized for acute episodes of disease [9,10,21].

Home appears as one of the most important contexts for the provision of care, performed not only by health professionals, where the role of nurses stands out, but especially by informal caregivers [32]. However, the overload to which informal caregivers may be subjected during this process, which is not always short or linear, as well as the lack of support, has been shown to be a predictor of early institutionalization [21], highlighting the increased risk of depression, fatigue and burnout, summarizing factors which should be taken into account in the whole process of intervention, follow-up and monitoring, with the need for investment in training, involvement and support, not only for the person being cared for, but also for the informal caregivers [11,25,32,34].

### 4.5. Informal Caregiver

The loss of independence in performing daily life activities that ageing brings about leads the informal caregiver of the elderly person to change routines, lifestyles [35] and family dynamics [36], with negative consequences on his/her physical and mental health. Thus, the informal caregiver is often forced to abandon life projects, especially when witnessing the degradation of the cognitive capacity of the person receiving care [37]. The uninterrupted character that the role of informal caregiver forces in most situations, with an average of 15h of daily care provision, constitutes in itself a factor of overload and weariness [38].

Women are the most representative group of informal caregivers of older people [36,37], presenting a greater burden due to the accumulation of different social roles [35]; however, there are no differences regarding resilience between genders, greater resilience in caregivers of older people being displayed by children or spouses and those who are already retired [36].

The informal caregivers of the elderly person with Alzheimer’s recognize that the information they have in order provide care is, in most situations, insufficient for quality care, with inadequate care planning and lack of tools to deal with changes generated by the disease [39]. Thus, the relationship between the caregiver and the older person with Alzheimer’s may generate a dynamic that drives conflicts and tensions, directly affecting the person receiving care, the caregiver and family dynamics [39]. On the other hand, the caregiver of the elderly person recognizes that being a caregiver of his relative generates a strengthening of the affective relationship established [36].

Health care, in turn, should include assessments of physical, cognitive, affective, social, financial, environmental and spiritual components, with the purpose of establishing a tangible therapeutic plan and integrated multidisciplinary follow-up, maintaining functional capacity, social reintegration and maintenance, and promoting a decrease in institutionalization [40].

From this perspective, nurses emerge as the health professionals better positioned to fill in information gaps with the informal caregiver, as well as collaborate in adaptive strategies [35,39] through the analysis of the real needs of informal caregivers and can be seen as “social educators” [36] with consequent improvements in the quality of life of this binomial [38]. The preponderant role of transmission of knowledge and skills to the elderly person and the informal caregiver stands out, with a view to a safe transition and adaptation to the new health condition, giving continuity with the care initiated at hospital level, since health–disease transitions with less positive outcomes are generators of dependence [32]. The transition processes are then seen as complex, demanding and implying a paradigm shift in the provision of care to the vulnerable person, requiring reconfiguration of the nurse’s role and multidisciplinary conjugation, encompassing all participants in the process, not forgetting the informal caregiver [32,41].

Nevertheless, it is crucial to take into account that a large majority of informal caregivers of the elderly are themselves elderly and also need care [38]. The overload to which these caregivers are subject when performing their role as informal caregivers, often concomitant with their professions, are in themselves generating conflicts that result in the abandonment of care and inevitable institutionalization [9].

Thus, the paradigm of care focused on the elderly person encompasses biological and psychosocial aspects with the aim of a global and integrative perspective with the purpose of fully satisfying human rights, focusing on self-determination, valuing individual characteristics, expectations and potentialities, and emphasizing the quality of the relationship in the provision of care [32].

The goal of holistic, individualized and quality care provided to the elderly, family and/or informal caregiver contemplates room for a care partner, from the initial moment of diagnosis, involving multidisciplinary care, until evaluation, taking into account this multidisciplinary in the intervention plan, the nurse seen as an asset in providing family members and informal caregivers with better training and information, with a view towards the best and most adapted therapeutic process [42].

### 4.6. Limitations

The limitations of this research are fundamentally based on the fact that it is the first bibliometric analysis on the topic in question, and the amount of data to be extracted was quite large, leading to an exhaustive treatment.

Some publications may not have been analysed because they were not well catalogued or were not published according to the inclusion criteria. Other types of publication such as conference abstracts or books were not included in this analysis, which may have contributed to this limitation. More recent articles may not be included in the analysis, although they do not have a major impact on the final result.

Another limitation may be related to the fact that only one database was used, excluding SCI-E, SCOPUS, and EBSCO, among others.

## 5. Conclusions

This bibliometric review regarding the informal caregiver allowed us to conclude that there has been a growing interest in this topic by the scientific community, which has been more pronounced in the last 5 years and is usually associated with health areas, more specifically gerontology and mental illness.

In view of the expected increase in ageing and dependence rates in the coming years, it is expected that there will be a growing need for a transition element capable of responding to the needs of an increasingly elderly and dependent population with a view to maintaining quality of life, with the informal caregiver emerging as a partner in care, and adherence to the therapeutic regimen. It is then vital to understand the evolution of the concept itself, which has been suffering a lexometric changing over time, currently the term “informal caregiver” being the most commonly used and on which research tends to focus.

An evolutionary concept analysis according to Rodgers’ framework may point to directions to justify the lexometric changes detected.

Due to the great relevance and growing interest in the topic, this represents a concept that still requires further research, particularly regarding the concept definition and its role in today’s society as well as its impact on health–illness transitions or continuity of care. It is also concluded that the issues related to Alzheimer’s disease, elderly and institutionalization are closely associated with the concept of informal caregiver/family caregiver, which may be incorporated into future research.

## Figures and Tables

**Figure 1 jpm-12-00061-f001:**
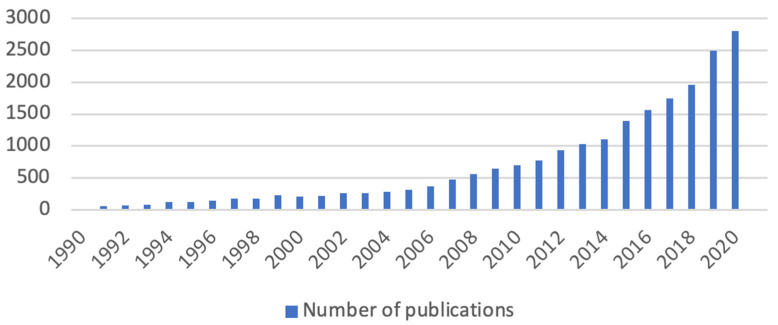
Annual number of publications on the informal caregiver (1990–2021); Source: elaborated by the author based on VOSviewer and CiteSpace data.

**Figure 2 jpm-12-00061-f002:**
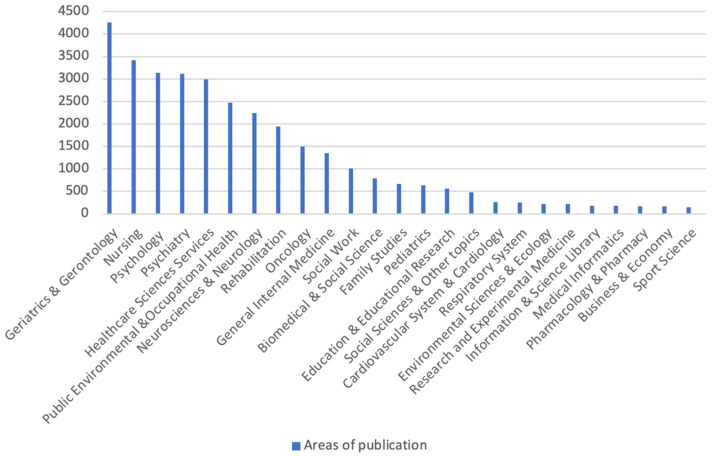
Number of publications by research area. Source: elaborated by the author based on VOSviewer and CiteSpace data.

**Figure 3 jpm-12-00061-f003:**
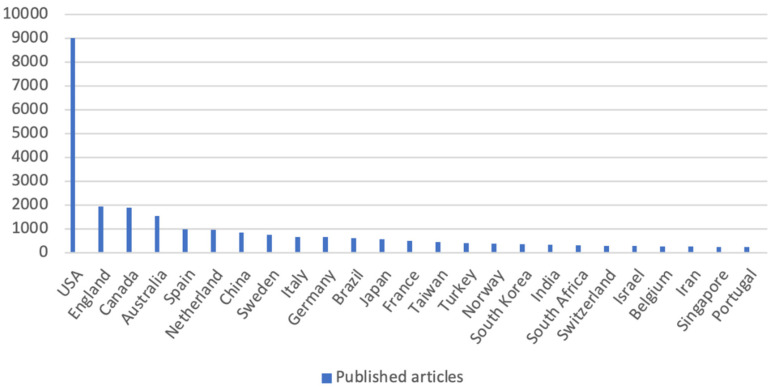
Number of published articles per country; Source: elaborated by the author based on VOSviewer and CiteSpace data.

**Figure 4 jpm-12-00061-f004:**
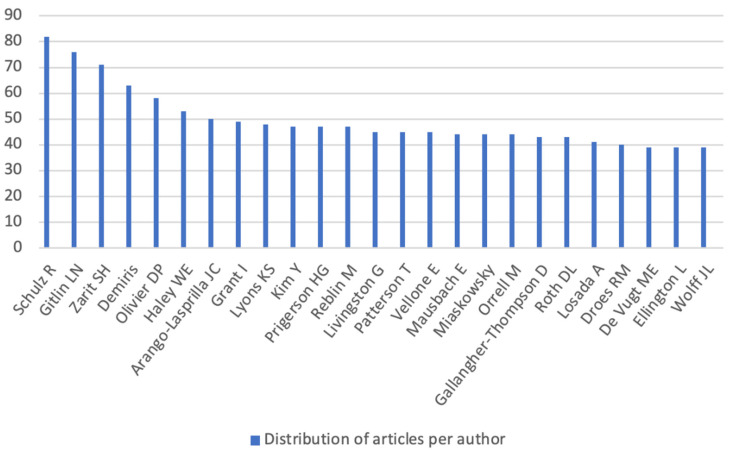
Distribution of articles per author. Source: Elaborated by the author based on VOSviewer and CiteSpace data.

**Figure 5 jpm-12-00061-f005:**
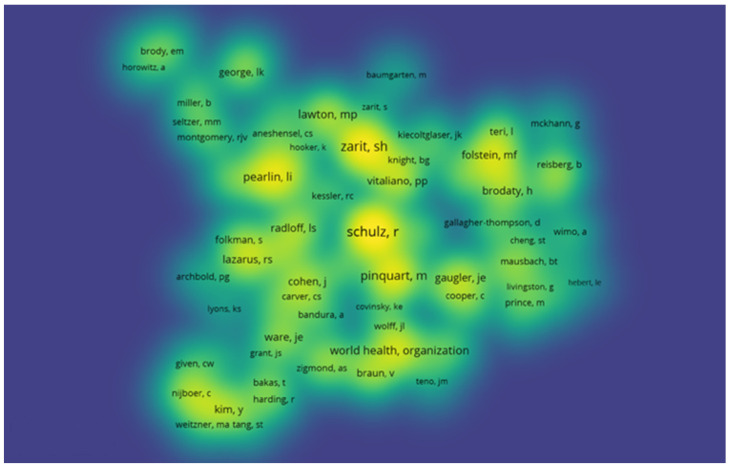
Density of author co-citations.

**Figure 6 jpm-12-00061-f006:**
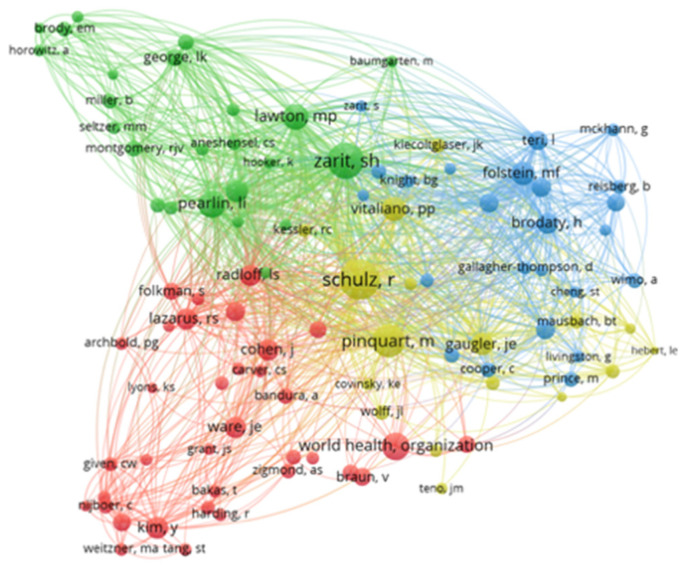
Authors’ co-citation network.

**Figure 7 jpm-12-00061-f007:**
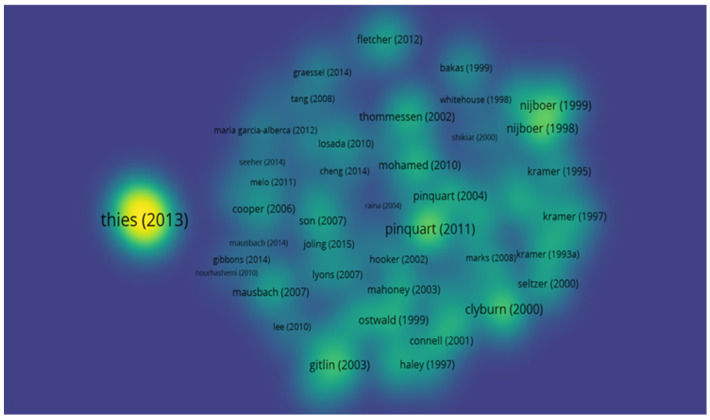
Density of bibliographic coupling.

**Figure 8 jpm-12-00061-f008:**
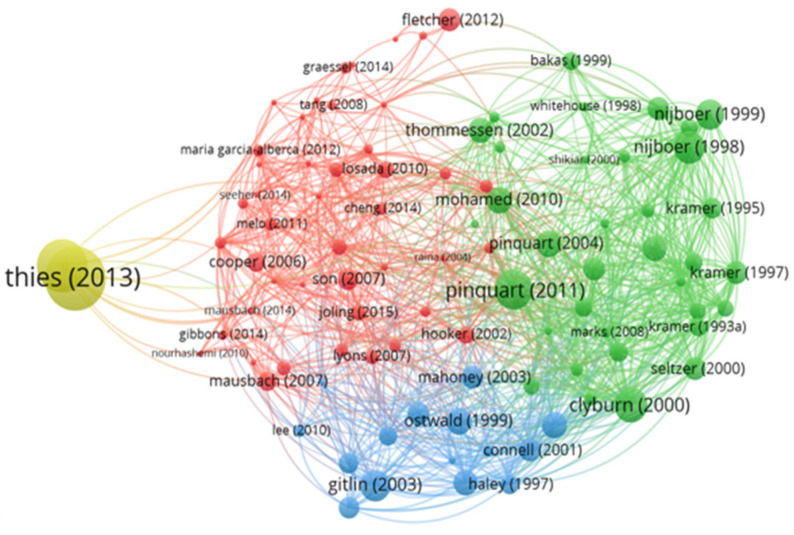
Bibliographic coupling network.

**Figure 9 jpm-12-00061-f009:**
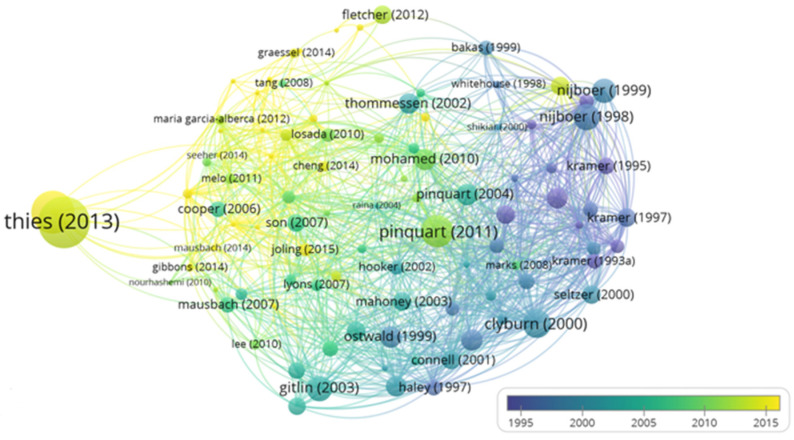
Chronological network of bibliographic coupling.

**Figure 10 jpm-12-00061-f010:**
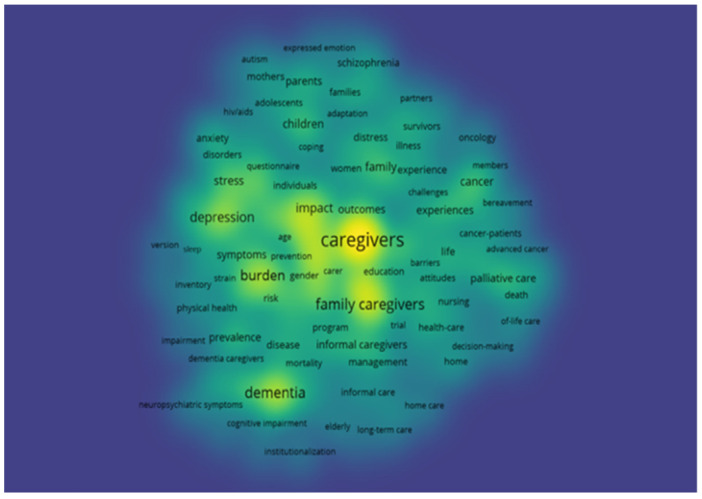
Co-occurrence density of the keywords.

**Figure 11 jpm-12-00061-f011:**
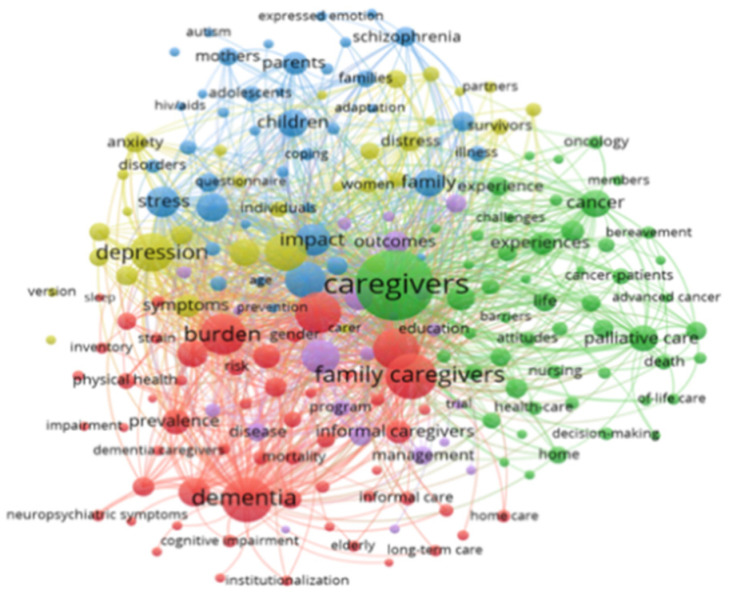
Co-occurrence network of the keywords.

**Figure 12 jpm-12-00061-f012:**
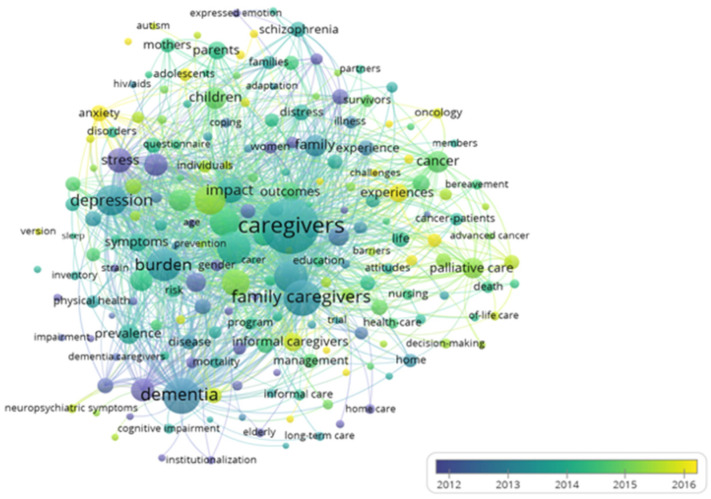
Chronological network of co-occurrence of keywords.

**Table 1 jpm-12-00061-t001:** Ranking of citation counts and centrality; Source: elaborated by the author based on VOSviewer and CiteSpace data.

Citation Counts	Centrality	References	Cluster
**1449**	1738	ZARIT SH, …, GERONTOLOGIST, 20	33.5
**1397**	1729	FOLSTEIN MF, …, J PSYCHIATRES, 12	37.5
**1296**	1723	PEARLIN LI, …, GERONTOLOGIST, 30	22.5
**1098**	1723	RADLOFF L S, …, APPLIED PSYCH MEASUREMENT, 1	35.5
**977**	1615	SCHULZ R, …, JAMA-J AM MED ASSOC, 282	13.5
**837**	1455	LAZARUS RS, …, STRESS APPRAISAL COP, 0	28.5
**796**	1453	BRAUN V, …, QUALITATIVE RES PSYC, 3	12.5
**781**	1432	PINQUART M, …, PSYCHOL AGING, 18	12.5
**629**	1388	ZIGMOND AS, …, ACTA PSYCHIAT SCAND, 67	32.5
**612**	1385	SCHULZ R, …, GERONTOLOGIST, 35	12.5

**Table 2 jpm-12-00061-t002:** Top 15 keywords with strongest citation bursts; Source: elaborated by the author based on VOSviewer and CiteSpace data.

Keywords	Year	Strength	Begin	End	1995–2025
Alzheimer’s disease	1995	54.33	1995	2025	▃▃▃▃▃▃▃▃▃▃
Family caregiver	1995	21.18	1995	2025	▃▃▃▃▃▃▃▃▃▃
Elderly	1995	21.02	1995	2025	▃▃▃▃▃▃▃▃▃▃
Institutionalization	1995	16.37	1995	2025	▃▃▃▃▃▃▃▃▃▃
Stress	1995	16.29	1995	2025	▃▃▃▃▃▃▃▃▃▃
HIV/AID	1995	13.68	1995	2025	▃▃▃▃▃▃▃▃▃▃
Assessment	1995	11.77	1995	2025	▃▃▃▃▃▃▃▃▃▃
Elder care	1995	11.76	1995	2025	▃▃▃▃▃▃▃▃▃▃
Ethnicity	1995	10.23	1995	2025	▃▃▃▃▃▃▃▃▃▃
Alzheimer disease	1995	9.93	1995	2025	▃▃▃▃▃▃▃▃▃▃
Aging	1995	8.32	1995	2025	▃▃▃▃▃▃▃▃▃▃
Support group	1995	7.28	1995	2025	▃▃▃▃▃▃▃▃▃▃
African American	1995	6.76	1995	2025	▃▃▃▃▃▃▃▃▃▃
Respite	1995	6.14	1995	2025	▃▃▃▃▃▃▃▃▃▃
Women	1995	5.84	1995	2025	▃▃▃▃▃▃▃▃▃▃

The red line represents the time line related to the keywords with strongest citation bursts.

**Table 3 jpm-12-00061-t003:** Top 15 references with strongest citation bursts; Source: elaborated by the author based on VOSviewer and CiteSpace data. (All links in this table have been accessed on the 25 June 2021.)

Keywords	Year	Strength	Begin	End	1995–2025
GEORGE LK, 1986, GERONTOLOGIST, V26, P253, https://doi.org/10.1093/geront/26.3.253	1986	158.39	1995	2025	▃▃▃▃▃▃▃▃▃▃
Braun V, 2006, QUALITATIVE RES PSYC, V3, P77, https://www.tandfonline.com/doi/abs/10.1191/1478088706qp063oa	2006	145.1	2006	2025	▂▂▂▂▂ ▃▃▃▃▃
STONE R, 1987, GERONTOLOGIST, V27, P616, https://doi.org/10.1093/geront/27.5.616	1987	139.14	1995	2025	▃▃▃▃▃▃▃▃▃▃
Adelman RD, 2014, JAMA-J AM MED ASSOC, V311, P1052, https://doi:10.1001/jama.2014.304	2014	95.38	2014	2025	▂▂▂▂▂ ▃▃▃
CANTOR MH, 1983, GERONTOLOGIST, V23, P597, https://doi.org/10.1093/geront/23.6.597	1983	91.62	1995	2025	▃▃▃▃▃▃▃▃▃▃
SCHULZ R, 1995, GERONTOLOGIST, V35, P771, https://doi.org/10.1093/geront/35.6.771	1995	89.76	1995	2025	▃▃▃▃▃▃▃▃▃▃
ZARIT SH, 1986, GERONTOLOGIST, V26, P260, https://doi.org/10.1093/geront/26.3.260	1986	81.06	1995	2025	▃▃▃▃▃▃▃▃▃▃
BRODY EM, 1985, GERONTOLOGIST, V25, P19, https://doi.org/10.1093/geront/25.1.19	1985	69.16	1995	2025	▃▃▃▃▃▃▃▃▃▃
POULSHOCK SW, 1984, J GERONTOL, V39, P230, https://doi.org/10.1093/geronj/39.2.230	1984	64.39	1995	2025	▃▃▃▃▃▃▃▃▃▃
Tong A, 2007, INT J QUAL HEALTH C, V19, P349, https://doi.org/10.1093/intqhc/mzm042	2007	61.1	2007	2025	▂▂▂▂ ▃▃▃▃
DEIMLING GT, 1986, J GERONTOL, V41, P778, https://doi.org/10.1093/geronj/41.6.778	1986	59.18	1995	2025	▃▃▃▃▃▃▃▃▃▃
LAWTON MP, 1969, GERONTOLOGIST, V9, P9, https://doi.org/10.1093/geront/9.1.9	1969	58.41	1995	2025	▃▃▃▃▃▃▃▃▃▃
COLERICK EJ, 1986, J AM GERIATR SOC, V34, P493, https://doi.org/10.1111/j.1532-5415.1986.tb04239.x	1986	57.51	1995	2025	▃▃▃▃▃▃▃▃▃▃
CHENOWETH B, 1986, GERONTOLOGIST, V26, P267, https://doi.org/10.1093/geront/26.3.267	1986	57.35	1995	2025	▃▃▃▃▃▃▃▃▃▃
Mittelman MS, 1996, JAMA-J AM MED ASSOC, V276, P1725, http://doi:10.1001/jama.1996.035402100330	1996	56.38	1996	2025	▂ ▃▃▃▃▃▃▃▃▃

The red line represents the time line related to the references with strongest citation bursts.

**Table 4 jpm-12-00061-t004:** Silhouette indexes by cluster, year and size; Source: elaborated by the author based on VOSviewer and CiteSpace data.

Cluster	Silhouette Index	Mean (Year)	Size
**0**	0.829	1990	746
**1**	0.771	2006	649
**2**	0.809	2005	517
**3**	0.782	2001	409

## Data Availability

Not applicable.

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
