# Peer review of "Bibliometric Analysis of the Informal Caregiver’s Scientific Production"

_jpm, 2022, doi:10.3390/jpm12010061_

Round 1
Reviewer 1 Report
Dear Authors,
Below you can read my comments after carefully reviewing your paper:
- The manuscript is very well written: clear sentences and good punctuation.
- I evaluate positively that the search has not been very restrictive, and that a lot of information was included in a broad chronological review.
- In my opinion, the title does not obey the content of the article. Your work presents not the evolution of the concept, but the evolution of the articles published on this issue. As you point out in the Introduction, you present the characteristics of the scientific production on the topic.
- Keywords are not really representative of the study. For example, I would remove concept formation (this topic has not been covered throughout the article) and add informal caregiver (in addition to caregiver).
- Lines 26-27. I do not believe that this divergence between countries and the data provided below is relevant to the issue of caregivers. The data is the increase in average life expectancy, and the increase in care needs. In fact, this data about discrepancy is not discussed later.
- Lines 55-56. Individual needs of each person are redundant.
- Line 102. From my point of view, bibliometric analysis is not “a relatively new methodology”. There are already classic bibliometric articles from the last century.
- How has Figure 3 been prepared? There is not a clear description of that. Has each of the articles been categorized one by one to make this figure, or has it been done with the programs mentioned above? I would ask for details on how the Figures were obtained.
- Starting from section 3.3, general and Portuguese data are given. Why point to specific data for Portugal, and not for the UK, for example? It would be necessary to justify why. If the journal was Portuguese, or if there were any specificities in that country…
- To facilitate the reading of the paper, I would move some Figures and Tables to a section of Annexes, and I would only give the relevant information in the body of the manuscript. At the discretion of the authors and what information they consider most important, they would choose what to move and what to keep in the original manuscript. I would recommend choosing only the most powerful graphic representations.
- Given the results, what can we say about the research question? They are descriptive data, but what is the really important contribution?
- Lines 413-414: It is hard to understand how from the analysis of Table 1, it is possible to state that: “...the most relevant research frontiers are "Alzheimer's Disease", "Elderly" and "Institutionalization". A justification is necessary.
- The three Sections in the Discussion about Alzheimer’s disease, the elderly and institutionalization are not clearly linked to the Results. What is derived from the work done by the authors, and what is a compendium of information from other authors? The authors included are adequate, but what does this paper add as really new?
- Lines 442-450: The findings of this paragraph are not familiar to me. Have we described them previously, in the Results section?
- In the Conclusion section you say: “It is then vital to understand the evolution of the concept itself, which has been changing over time, and currently the term "informal caregiver" is the most commonly used and on which research tends to focus.” (Lines 575-577). This statement only explains a lexicometric change, but what about the conceptual implications?
- The Discussion section should discuss the implications of the findings in context of existing research, but the link between them is not explicit. In fact, many findings we have seen in the Results section are not discussed. You should choose the results that are important and “discuss” their implications.
Author Response
Dear reviewer, we would like to recognize you in advance for the thorough and rigorous review of our work, it is a very detailed and rigorous review that serves not only to enhance the article but above all as a fabulous learning opportunity.
Thank you once again for the feedback, it was crucial for us.
We would like to answer all your suggestions by the same order, starting by the title: we agree with your opinion, and decided to change the article name to: Bibliometric analysis of the Informal Caregiver scientific production;
Regarding the Keywords, we followed your suggestion as well, we removed the “concept formation” and add “informal caregiver”, leaving as final Keyword: informal caregiver; caregiver; bibliometrics; data analysis;
Lines 26-27 were removed as they effectively did not add any information that would be later on discussed or linked to any of the subjects;
Line 55-56, was a spelling error, “of each person” was deleted from the article;
Line 102, we agree with your suggestion, since 1970 it might not be sensible to define as something relatively new, changing on the paper to: “As this methodology has been rapidly growing academic interest among researchers due to the increased availability of databases and software used in the methodology.”
Figure 3, as well as all the others were elaborated by us in the form of figures or tables, having in consideration all the data extracted from both computer programs: VOSViewer and CiteSpace. We also add to the figure’s legend, “based on CiteSpace and VOSViewer data”;
This is a very well spotted point, in fact, as can be seen on the front page, all the authors work in Portugal, being the reality with easier access to us, however, the data extraction was based on 4 languages (English, French, Spanish and Portuguese), from our point of view was important to highlight the scientific production with least publications as well.
Regarding the figures, we appreciate your kind review and agree with it, as easy the reading of the paper and make it less distracting. In order to achieve a “cleaner” view, we opted to remove figure 2 and 4 regarding top 25 journal/magazines and distribution of articular per language respectively, as you highlighted, is all written on the paper and there is no need to repeat information.
Thank you for this suggestion in specific, it enhance the quality of the paper and was a good learning curve for us. To answer this suggestion we would like to relate it with the penultimate and last suggestions. Regarding the discussion, we rearranged the paragraphs and add some information, being the most important information in these 3 paragraphs:
“The largest number of publications appears in journals or magazines dedicated to Gerontology & Geriatrics, Nursing and Psychology. The largest number of co-citations belongs to Schulz, Gitlin and Zarit, being Zarit the author the author with the most published articles on the topic, followed by Folstein and Pearlin, appearing "Alzheimer´s disease", "institutionalization" and "family caregiver" as the keywords with the strongest citation burts. With support of table 2, it’s possible to indicate that these keywords appear with strongest citation bursts since the beginning of the 90's until current days, suggesting that they are themes that are still discussed by the authors. Eom & Fortunato (2011) [21], when studying the dynamic properties of citation flows, found that the first years after the publication of articles are characterized by citations bursts that work as an indicator of the popularity dynamics of the evidence produced and that have different durability over time.
The most commonly used terms in the publications are "caregivers" and "family caregivers", which appeared in 2013, with the term "informal caregivers" gaining greater relevance from 2016 onwards, maintaining its expression to date.
Given this lexometric change, and in order to avoid ambiguity in the applicability of the concept, it will be important to analyze it in a future study. According to Tofthagen & Fagerstrøm (2010) [22], nursing research should focus on the unambiguous use of con-cepts, for which Rodgers' method is a possible method.”
On the conclusion we also add some information regarding the change in lexicometric.
Regarding lines 413-414, was a spelling mistake and we appreciate the attention to the minimum detailed. The information supporting the search frontiers is on table 2, not 1, as the strongest keywords bursts citations are “Alzheimer’s disease; Elderly and Institutionalization”.
Regarding the discussion about Alzheimer’s disease; Elderly and Institutionalization, we tried to relate to the results by adding the information above, with the studies of Eom & Fortunato (2011), when studying the dynamic properties of citation flows. As article, is intended to add a view of the evolution of the scientific production, based on the literature and open the field for new investigations, namely the definition of the concept, incorporating the keywords detected with strongest citation bursts.
With concern to line 442-450, we did not find any familiarity, which we apologies in advance, if you could highlight what the similarity is, we would be very much appreciative.
The last 2 suggestions we tried to answer above, but if any suggestion on question, please fell free.
Once again, thank you very much for your review, was very productive and the article gain much quality with your suggestions which we appreciate, giving us the opportunity to improve our paper but ourselves too.
Kind regards,
The authors
Reviewer 2 Report
The authors present a bibliometric analysis of the concept of the informal caregiver and its habitual establishment and the challenges generated by caring for the sick, and the impact on health on the informal caregiver. Next, I will go on to indicate some aspects of improvement for the authors to make the indicated corrections and observations / suggestions offered by the reviewer:
The numbers of the references in text appear next to the final word and the authors must leave a short space.
Figures
In relation to figure 2, indicate to the authors that there is text cut in some boxes and that it would be advisable to put in notes, if the colors follow any criteria (put the color, and indicate orange: Nursing, brown = xxx)
On the other hand, these figures could be improved or replaced by a system of tables that collect this information.
Figure 2 could be eliminated since this information is collected in text and is repeated.
Figure 4 does not provide any information. It is already indicated in the text. I recommend withdrawing it or referring to this topic by presenting the data, but referring to the figure without writing all the information.
Figure 6 also does not provide any new information, except for the authors who appear for the first time in the figure. Do not repeat the information in text and refer to the figure or translate all the information into text. These are the two options. Also, they re-cut the text in the boxes.
On line 231 you could remove the initials of the authors and leave only the last name. The same thing happens in the pictures.
In line 248 the authors have put a reference number 17 in small size. This must be corrected.
In lines 250 and 251 they must remove the initials of the authors' names and also, in following. Do the same where this occurs throughout the manuscript.
In line 267 a reference appears again with the size of the number 16 in small
Table 1 In the column of references they must follow the same criteria for all. If you put them abbreviated, they should do so in all of them. Since it puts some abbreviated and others not like "Applied Psychologist Measurement" (you must abbreviate it)
From line 283, authors' names appear, but without putting them according to the citation style correctly. For example, by Thies, 2013, put by Thies (2013) and do the same with the following.
On line 287 there is a dot in front of “Gitlin” that you must remove From line 355, remove the initials of the authors' names again With respect to the figures, the final point must be removed from the titles (eg Figure 10 and others. In table 2, Alzheimer's disease lacks an umlaut.
On line 447 they have written the word "undoubtedly" in a different style. Must put the same.
On line 493 when you say "highlighting the increased risk of depression" I recommend that you provide the reference of the article: - Camacho-Conde JA, Galán-López JM. Depression and Cognitive Impairment in Institutionalized Older Adults. Dement Geriatr Cogn Disord. 2020; 49 (1): 107-120. doi: 10.1159 / 000508626. which also illustrates and supports what it says here
Author Response
Dear reviewer, we would like to express our recognition for your thorough and rigorous review of our paper, your suggestions were taken in account and from our perspective enhanced the document and gave us a crucial learning opportunity which we would like to thank in advance.
We will try to answer all the suggestions following the same order that they were presented.
The numbers of the references were spaced as suggested, was a format error from us;
All the figures with incomplete words were replaced for a figure in form of bars graphic, which is easier to interpret and have all the information for the reader. The precious figure was generated from the computer program used to extract data, not having any rational with regards to color or form of the spaces, but with a bar graphic it´s easier to understand the information we are trying to transmit, which we thank you for the suggestions as the paper became user friendly to read and interpret;
Figure 2 was eliminated as suggested, as all the information is already in the text as the reviewer very well highlighted;
Figure 4 was also removed from the paper, as all the information is on the text;
Regarding figure 6, we opt to keep it as show some authors that are not mentioned and might be useful as curiosity, but we had your suggestion in account, agreeing that the information in the text is duplicate and was deleted, leaving only the crucial one;
Line 231, as well as the rest of the paper, we appreciate the review, aggreging with your suggestion and removing the authors initials, leaving last names only;
Line 267, was a format error which was promptly corrected;
Table 1, we add in account your suggestion and abbreviated the names, as we agree that would not bring any useful information related to the paper if decided to keep the full name;
Regarding the correct style of citation, we appreciate the detailed review, and agree with your suggestion, correcting the citation style and applying to the whole article;
Concerning line 287, the format error was corrected as your suggestion, as well as line 355 which the initials were removed, leaving the last names only, deleting the full stop signs on the end of the figures and tables legends;
Line 447, was once again a format error, which we appreciate your detailed review, being corrected as very well suggested from the reviewer;
As the last review, we would like to recognize your suggestions, as we agree to add to the references as will enhance the quality of the article, becoming an added value to it, illustrating perfectly the idea we were trying to transmit.
Once again we would like to thank you for such a professional and detailed review, was very educative and added value to the article as a document.
Kind regards,
The authors